cognition/psychology

ageing, visual search task, facial expression detection, normal facial expression, anti-facial expression

**Authors for correspondence:**
Akie Saito
e-mail: akie.h.saito@gmail.com
Wataru Sato
e-mail: sato.wataru.4v@kyoto-u.ac.jp

# Older adults detect happy facial expressions less rapidly

Akie Saito, Wataru Sato and Sakiko Yoshikawa

Kokoro Research Center, Kyoto University, 46 Shimoadachi, Sakyo, Kyoto 606-8501, Japan

 AS, 0000-0002-6961-2932; WS, 0000-0002-5335-1272; SY, 0000-0002-0122-2006

Previous experimental psychology studies based on visual search paradigms have reported that young adults detect emotional facial expressions more rapidly than emotionally neutral expressions. However, it remains unclear whether this holds in older adults. We investigated this by comparing the abilities of young and older adults to detect emotional and neutral facial expressions while controlling the visual properties of faces presented (termed anti-expressions) in a visual search task. Both age groups detected normal angry faces more rapidly than anti-angry faces. However, whereas young adults detected normal happy faces more rapidly than anti-happy faces, older adults did not. This suggests that older adults may not be easy to detect or focusing attention towards smiling faces appearing peripherally.

## 1. Introduction

The ability to detect emotional facial expressions is essential to understanding the feelings of others and enjoying successful social interactions. Earlier psychological studies using visual search paradigms reported that young adults detect emotional facial expressions (e.g. angry or happy faces) more rapidly than neutral facial expressions [1–5]. For instance, in Williams *et al.*'s [5] experiment, participants detected one different facial expression in a line-up of facial expressions. The reaction time (RT) for identifying sad or happy expressions among several distractors (neutral expressions) was shorter than the RT for detecting neutral faces among emotional faces. Such rapid detection of emotional facial expressions is attributable not to the visual properties of the faces *per se* but rather to the emotional significance of such expressions. This interpretation was supported by a study that compared the speed of detecting normal angry and happy facial expressions to that of detecting the corresponding 'anti-expressions' in several neutral faces [2]. Anti-expressions were created via computer morphing, which imparted the same degree of visual change to neutral facial expressions but in the opposite direction of normal emotional expressions [6]. In general, anti-expressions are created by reversing the directions of the facial characteristic of emotional expressions but retaining the general

facial configuration. If an angry face features a V-shaped eyebrow and a neutral face a horizontal eyebrow, the computer generates an anti-expression with the eyebrows in an upside-down V ($\Lambda$) shape. Anti-expressions do not convey specific emotions and are typically recognized as emotionally neutral [6]. In this sense, anti-expressions serve as emotionally neutral facial stimuli while controlling for visual facial characteristics. The RTs for detecting normal angry or happy faces were shorter than the RTs for detecting the corresponding anti-expressions. As emotional expressions and anti-expressions both lie equidistant from neutral facial expressions, these results showed that the emotional significance of faces, not their visual properties *per se*, contribute to the rapid detection of such expressions by young adults.

Detecting emotional facial expressions has also been examined in a visual search paradigm in older adults [7–9]. Such studies are particularly important; some older adults do not reliably recognize certain emotional facial expressions (such as negative expressions associated with anger) [10–19]. It is possible that the detection of emotional facial expressions precedes conscious recognition of the emotions imparted. Hence, losing one's ability to detect negative facial expressions (such as angry faces) may impair emotional recognition. Previous works compared the abilities of younger and older adults to detect emotional facial expressions [7–9]. Specifically, using a schematic face, Mather & Knight [8] demonstrated that older adults detected angry faces more rapidly than happy or sad faces. Hahn *et al.* [7] compared the attentional shifts of younger and older adults toward schematic emotional faces placed among neutral distractors; the RTs for locating angry faces on a screen were shorter than those for locating happy faces, which supports the data of Mather & Knight [8]. Although overall detection speed was slower among older than younger adults, a similar pattern emerged in both groups. The authors concluded that both older and younger adults efficiently detect angry faces. This was confirmed using photographs of real faces; Ruffman *et al.* [9] reached similar conclusions.

However, the conclusion that older adults effectively detect angry faces was derived by comparing the speeds for detecting angry, happy and sad facial expressions [7–9] rather than by directly comparing the RTs to those for detecting emotionally neutral facial expressions. Thus, it remains unclear whether older adults indeed detect angry (compared to neutral) facial expressions as efficiently as young adults. The intact automatic detection hypothesis for older adults will be validated only if older participants detect angry facial expressions as rapidly as young adults.

Moreover, given the lack of comparisons between happy and emotionally neutral facial expressions, it remains unclear whether older adults detect happy facial expressions more rapidly than neutral facial expressions. Young adults do so, but visual search tasks exploring the capacity of older adults have not been performed. The question is interesting, in the sense that exploring the effects of happy (positive) stimuli on detection speed might enhance understanding of how positive emotions and attention interact, a topic that has received increasing attention in recent years. Recent studies on the relationship between emotion and attention in young adults have revealed that both positive and negative stimuli are automatically processed and rapidly capture attention [20–23]. Accumulating evidence suggests that the attention-capturing power of positive stimuli may be even stronger than that of negative stimuli when emotional stimuli are presented as distractors while participants engage in tasks that demand attention [22,24,25]. For instance, using positive and negative faces as distractors, Gupta *et al.* [22] found that only positive distractors compromised participant performance in a high-load letter-search task, although both positive and negative faces distracted participants under low-load conditions. This is explained by the nature of positive stimuli, which readily capture attention and are difficult to ignore even in resource-demanding situations, because the attentional resources required to note positive stimuli are less than those required to recognize negative stimuli [22,25]. This is supported by various studies that have explored the processing of positive stimuli using various methods [24,26,27]. These findings are not incompatible with those of the visual search studies reporting the rapid detection of positive (happy) facial expressions by young adults. If attention is automatically paid to both negative and positive information during later adulthood, older adults will also detect happy facial expressions more rapidly than emotionally neutral expressions.

Here we explored this topic by comparing the RTs for detecting normal angry and happy facial expressions (compared to the corresponding anti-expressions placed within a series of neutral facial expressions) between younger and older adults. We used a visual search task (figure 1). As mentioned previously, anti-expressions serve as emotionally neutral facial stimuli while controlling for visual facial characteristics [2,6]. Thus, comparing the RTs for detecting such expressions and normal emotional facial expressions would clarify whether the emotional significance of facial expressions aids in the rapid detection of such expressions by older adults. We compared the RTs for detecting each type of normal expression and anti-expression in both age groups using the methods we used previously [2]. If both groups shared the same pattern of detecting emotional facial expressions and anti-expressions, the intact

**Figure 1.** Illustrations of stimuli presented (*a*) and the visual search display (*b*). In the actual experiment, the stimuli were composed of photographs of real faces.

**Table 1.** Mean (with standard deviation) demographic data for young and older participants.

|  | young | older |
|---|---|---|
| age | 21.4 (2.0) | 72.0 (5.5) |
| years of education | 15.1 (1.8) | 13.3 (2.4) |
| digit-span backward[a] | 9.9 (2.8) | 6.6 (1.8) |
| knowledge[a] | 20.2 (3.0) | 17.6 (5.0) |
| depression[b] | 6.2 (6.5) | 10.4 (6.2) |

[a]From Wechsler Adult Intelligence Scale III.
[b]From Beck Depression Inventory II.

automatic detection hypothesis would be supported in older adults. By contrast, if older adults did not detect emotional facial expressions more rapidly than the corresponding anti-expressions, automatic detection of emotional expressions would be impaired. We also assessed subjective valence and arousal ratings for each target stimulus and used the data to assess emotional impact.

# 2. Material and methods

## 2.1. Participants

Thirty young adult participants (17 females and 13 males, mean ± s.d. age = 21.4 ± 2.0 years), all of whom were either undergraduate or graduate students at Kyoto University, were recruited through advertisements for a temporary job on campus and compensated for their time. Thirty-two older participants (16 females and 16 males, mean ± s.d. age = 71.9 ± 5.5 years) were recruited from a local senior human resource centre in Kyoto and were also paid for their participation. The required sample size for a repeated measures analysis of variance (ANOVA) with one between- and one within-subjects factor (two levels each; assuming the analysis of RT differences between normal expression and anti-expression targets) was determined via *a priori* power analysis using G*Power (v. 3.1.9.2) [28] assuming an $\alpha$ level of 0.05, a power $(1 - \beta)$ of 0.80 and a repeated measures correlation of 0.2 (estimated based on our previous data). As the effect sizes were unclear, we predicted medium effects ($f = 0.25$). The power analysis showed that at least 54 participants were required. All were Japanese. Older subjects were screened for dementia with a Japanese version of the mini-mental state examination [29]. None of the older participants scored below the cut-off point of 24 (mean ± s.d. score = 28.7 ± 1.3). All of the participants were right-handed, which we confirmed based on the Edinburgh Handedness Inventory [30]. Although additional volunteers participated, their data were excluded because of left- or bi-handedness (one female and four males in the young group, one female and one male in the older group). We summarize the demographic data for both age groups participating in the experiment in table 1.

The participants reported having normal to corrected-to-normal vision. No participant had any history of neurological or psychiatric disorders or was taking medication for such disorders. After the experimental procedure was explained, all participants gave written informed consent.

## 2.2. Design

The experiment featured a three-factor mixed design, with group (young, older) as a between-participants factor and stimulus type (normal expression, anti-expression) and emotion (anger, happiness) as within-participants factors.

## 2.3. Apparatus

The stimuli for all tasks were presented on a 19 inch monitor (HM903D-A, Iiyama, Tokyo, Japan) with a refresh rate of 150 Hz and a resolution of $1024 \times 768$ pixels, which was controlled by Presentation 14.9 (Neurobehavioral Systems, San Francisco, CA, USA) and connected to a Windows personal computer (HP Z200 SFF, Hewlett-Packard, Tokyo, Japan). Responses were acquired via a response box (RB-530, Cedrus, San Pedro, CA, USA).

## 2.4. Stimuli

Photographs of real faces displaying normal expressions and anti-expressions of anger and happiness served as target stimuli; neutral facial expressions served as distractor stimuli. The stimuli were those of Ekman & Friesen [31] and thus identical to those used in the previous studies [2,32,33]. Each face subtended a visual angle of 1.8° horizontally and 2.5° vertically. The normal angry and happy facial expressions and the neutral expressions (distractors) were selected from a facial expression database of grey-scaled photographs of one female (PF) and one male (PE) model (both Caucasian) featuring either angry, happy or neutral expressions. Two photographs of the same model (either sex) served as target stimuli and one (either sex) served as a distractor stimulus. Photographs showing bared teeth were not used. No participant was acquainted with either model.

We created anger and happy anti-expressions by modifying the neutral facial expressions of the two models; these anti-expressions also served as target stimuli.

We created anti-expressions from normal expressions using computer-morphing software (FUTON System, ATR, Soraku, Japan). First one author manually identified the coordinates of 79 facial points and realigned them based on the coordinates of the bilateral irides. Next the distances between the feature points of the emotional and neutral facial expressions were calculated, and the anti-expression feature points positioned by moving each point of either neutral expression the same distance, but in the opposite direction, to that of the corresponding point of the emotional face [6].

To eliminate any possible effects of contours or visible hairstyles on detection processing, we cropped all of the photographs into an oval slightly within the frame of the face using Photoshop 5.0 (Adobe, San Jose, CA, USA), which we also used to make minor adjustments to colour, by a few pixels, and contrast between light and shade.

Eight possible positions for the presentation of facial stimuli were prepared, each being separated by 45° and positioned in a circular configuration ($10.0° \times 10.0°$). The experimental stimuli were presented occupying four of the eight possible positions, of which two stimuli were presented on the left side of the screen and the remaining two were presented on the right side (as illustrated on the right-hand side of figure 1). Every combination of the four positions was presented an equal number of times. The positions of the target stimuli were selected in a pseudo-random manner for the target-present trials, such that they appeared on the left side of the screen in half of the trials and on the right side in the remainder. In the target-present trials, one face from among the target stimuli was presented together with three identical neutral faces. In the target-absent trials, all faces were neutral.

## 2.5. Procedure

The participants performed the visual search task and then completed the rating task in a soundproofed room (Science Cabin, Takahashi, Kensetsu, Tokyo, Japan). They were asked to sit in a chair, keeping their chins fixed in a steady position at a distance of 80 cm from the monitor screen.

The participants were instructed to indicate whether a different face was present or whether all faces were identical in each stimulus array of four faces as quickly and accurately as possible by pressing the

assigned button on the response box using their right and left index fingers. In addition, they were asked to focus their eye gaze on a fixation cross ($0.9° \times 0.9°$) positioned at the centre of the display while holding their index fingers on the two response buttons. The participants completed 36 practice trials, which were followed by the main trials.

Each trial began with the presentation of a fixation cross for 500 ms, followed by a stimulus array of four faces. The faces remained visible until the participants pressed a button, and then a new trial was initialized by the reappearance of the fixation cross. The experiment consisted of four blocks of 72 trials (288 trials in total), with equal numbers of target-present and target-absent trials. Each target type was presented the same number of times within each block (nine times for each target type, for a total of 36 times). The trials were presented pseudo-randomly so that no identical targets appeared in the same positions in successive trials. The assignments of the response buttons were counterbalanced across participants.

The rating task followed the visual search task. In this task, the participants rated the facial stimuli that had been presented as targets (eight photographs) and distractors (two photographs) in the previous task. Each face was presented one at a time, and the participants were asked to rate how they felt while seeing each facial stimulus and to evaluate the stimuli based on their feelings, in terms of intensity of arousal and emotional valence, on a nine-point scale ranging from 1 (low arousal or negative for either arousal or valence rating) to 9 (high arousal or positive for each rating). Nearly half of the participants within each age group completed arousal ratings for stimuli first, which were followed by the valence ratings. The remaining participants rated the arousal and valence in the reverse order. The stimuli were presented in random order.

## 2.6. Data analysis

SPSS 16.0 J software (SPSS Japan, Tokyo, Japan) was used to perform statistical analyses. The $\alpha$ level was set to 0.05.

The mean RTs for the correct responses for each condition of the target-present trials were calculated, excluding measurements ±3 s.d. from the mean for each participant as artefacts (1.7% of responses). The data were subjected to a log transformation to satisfy normality assumptions for the subsequent analyses. Mean log-transformed RTs were analysed via a three-way repeated measure ANOVA with age group (young and older) as a between-participants factor and stimulus type (normal expression and anti-expression) and emotion type (anger and happiness) as within-participants factors. Follow-up analyses for simple effects were performed when a significant three-way interaction was found. If there was a significant highest order interaction, other effects or interactions were not subjected to interpretation as they would be qualified by the highest order interaction. In preliminary analyses, we performed four-way ANOVAs on the log-transformed RTs adding either participant sex, the visual fields of the presented faces or the presentation block. We found no significant four-way interaction; in other words, we detected no moderation of our effect-of-interest three-way interaction ($F < 1.44$, $p > 0.10$). Accordingly, we report only the results of the aforementioned three-way ANOVA. The accuracy and valence and arousal rating scores were analysed in the same manner as the RTs.

## 3. Results

### 3.1. Visual search task

Figure 2 shows the mean RTs in each age group for each of the target conditions. The results of the three-way ANOVAs showed a significant three-way interaction, which suggested that speeds for detecting normal facial expressions versus anti-expressions varied with respect to both age group and emotion type ($F_{1,60} = 5.34$, $p < 0.05$, $\eta_p^2 = 0.082$). In addition, a significant interaction was apparent between stimulus type and emotion type ($F_{1,60} = 49.61$, $p < 0.001$), as were significant main effects of age group, stimulus type and emotion type ($F_{1,60} > 23.56$, $p < 0.001$).

We then performed simple effect analyses. First we tested simple-simple main effects of stimulus type in the young group. The effects were significant for both angry and happy expressions ($F_{1,120} > 5.20$, $p < 0.05$), which indicated that young adults responded faster to normal angry and happy facial expressions than to the corresponding anti-expressions. By contrast, the older group exhibited a significant simple-simple main effect of stimulus type for angry expressions ($F_{1,120} = 65.44$, $p < 0.001$) but not for happy expressions ($F_{1,120} = 0.10$, $p > 0.10$). Hence, older adults more rapidly detected normal angry expressions than anti-angry expressions, but no significant difference in speeds for detecting happy

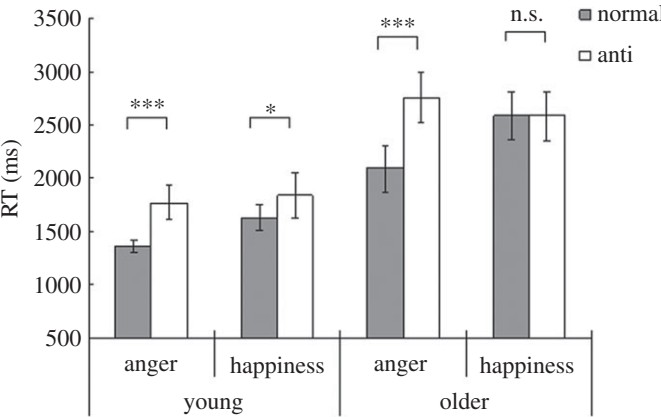

**Figure 2.** Mean reaction times (RTs) with standard error bars for each target condition in the young and older groups. Asterisks represent significant simple main effects of stimulus type (***$p < 0.001$; *$p < 0.05$; n.s., not significant).

**Table 2.** Mean (with standard error) subjective ratings of valence and arousal in the young and older groups.

| | valence | | | | arousal | | | |
| | normal | | anti- | | normal | | anti- | |
| | anger | happiness | anger | happiness | anger | happiness | anger | happiness |
|---|---|---|---|---|---|---|---|---|
| young | 2.5 (0.2) | 7.0 (0.2) | 4.8 (0.2) | 3.7 (0.2) | 6.5 (0.3) | 5.3 (0.2) | 4.3 (0.2) | 4.2 (0.3) |
| older | 2.9 (0.2) | 7.0 (0.2) | 5.5 (0.2) | 3.7 (0.1) | 5.1 (0.4) | 5.6 (0.3) | 4.5 (0.2) | 4.8 (0.3) |

expressions and anti-happy expressions was evident. Next we tested simple-simple main effects of emotion type. Significant effects for normal expressions were evident in both the young and older groups ($F_{1,120} > 26.35$, $p < 0.001$), indicating that both groups responded more rapidly to normal angry facial expressions than to normal happy expressions. The results of tests of simple-simple main effects of emotion type for anti-expressions showed that emotion type was significant only in the older group ($F_{1,120} = 5.39$, $p < 0.05$), which indicated that older adults exhibited a detection speed advantage for anti-happy over anti-angry expressions. Finally, tests of simple-simple main effects of age group confirmed that the effects were significant across all conditions ($F_{1,240} > 16.74$, $p < 0.001$), which revealed that older adults responded more slowly to the target stimuli than did young adults.

We analysed accuracy using a three-way repeated measures ANOVA in the same manner as we earlier evaluated the log-transformed RTs. No significant three-way interaction was apparent ($F_{1,60} = 0.93$, $p > 0.10$, cf. electronic supplementary material, table S1). We found no evidence of a speed/accuracy trade-off or a between-groups difference.

## 3.2. Rating task

Table 2 shows the subjective rating results for valence and arousal. These values were submitted to three-way ANOVAs with age group, stimulus type and emotion type as factors. In terms of valence ratings, no significant three-way interaction ($F_{1,60} = 0.26$, $p > 0.10$, $\eta_p^2 = 0.004$) or main effects of age group ($F_{1,60} = 2.70$, $p > 0.10$) were detected. Significant two-way interactions were observed between age group and emotion ($F_{1,60} = 8.24$, $p < 0.01$), and between stimulus type and emotion type ($F_{1,60} = 333.03$, $p < 0.001$). Significant main effects of stimulus and emotion type were also observed ($F_{1,60} > 17.34$, $p < 0.001$).

Analyses of the arousal ratings demonstrated significant three-way interactions ($F_{1,60} = 4.02$, $p < 0.05$, $\eta_p^2 = 0.063$) and two-way interactions between age group and stimulus ($F(1,60) = 11.33$, $p < 0.005$) and between age group and emotion ($F_{1,60} = 5.72$, $p < 0.05$). In addition, a significant main effect of stimulus type was apparent ($F_{1,60} = 64.46$, $p < 0.001$).

Next, we analysed the three-way interaction among the arousal ratings. Simple-simple main effects of stimulus type were significant for angry and happy expressions among young adults ($F_{1,120} > 10.18$,

$p < 0.005$), demonstrating higher arousal when encountering normal expressions compared to anti-expressions. In older adults, this effect was significant only for happy expressions ($F_{1,120} = 6.10$, $p < 0.05$), which indicated that participants experienced more arousal when presented with normal happy expressions than anti-happy expressions. Simple-simple main effects of emotion type attained significance only for normal expressions among young adults ($F_{1,120} = 9.76$, $p < 0.005$). This suggested that young adults felt more arousal when presented with normal angry than normal happy expressions. Finally, significant simple-simple main effects of age were evident only for normal angry expressions ($F_{1,240} = 11.77$, $p < 0.001$), suggesting that such expressions elicited higher arousal in young than older adults.

## 4. Discussion

Previous studies have shown that young adults detect angry and happy facial expressions faster than emotionally neutral expressions [2,32,34]. However, data on older adults were lacking. Here we directly compared the abilities of older and younger adults to detect normal facial expressions and anti-expressions. Younger adults detected normal angry and happy facial expressions more rapidly than their anti-expression counterparts, consistent with previous findings [2,32,34]. More important, we found that older adults detected normal angry facial expressions more rapidly than the corresponding anti-angry expressions; however, this was not the case for normal happy facial expressions compared to the corresponding anti-happy expressions. Older adults rapidly detected angry facial expressions but not happy expressions. Overall, older adults responded to stimuli more slowly than younger adults, as found in prior studies using visual search paradigms [7–9]. Processing speed slows in later adulthood [35–37].

We found the clear evidence of intact automatic detection of angry facial expressions in older adults; this robust ability remains unchanged during adulthood. Both age groups detected normal angry facial expressions more rapidly than normal happy expressions, consistent with earlier findings [7–9]. The rapid detection of threatening, angry faces enhances survival by allowing a person to avoid physiological and psychological harm [38]. It is reasonable to assume that superior detection of angry compared to happy faces reflects innate self-preservation.

By contrast, only older adults did not detect normal happy facial expressions more rapidly than anti-happy expressions, which suggest that any emotional significance of happy expressions is reduced in older adults. What might cause this? It has been suggested that high-level attention to positive stimuli may be explained by today's competitive and hedonistic society [21]. Gupta [21] argued that it is important to detect positive stimuli, as these impart clues as to how we should behave. The utility of positive stimuli (including happy faces) might be particularly important in young adults, who are assumed to want to build and expand social relationships (future investments) [39]. When establishing new relationships is important, it is natural to focus on happy faces, because these indicate how one might advance one's social career or social status. Conversely, if establishing new social relationships is not important, subjects may sometimes ignore happy faces. Ageing is characterized by declining health and shrinking social networks [40–42]. Happy faces may not motivate the aged; they do not impart useful life clues.

The arousal ratings confirm that anti-expressions serve as valid control stimuli for emotional facial expressions. The higher arousal ratings recorded by both age groups for normal facial expressions than for anti-expressions suggest that the participants found anti-expressions to be less emotionally evocative than normal facial expressions. Furthermore, the higher arousal ratings generated by facial expression stimuli among young adults appear to be related to their speedy detection of these stimuli; normal angry expressions were detected most rapidly and induced the highest arousal ratings, followed by normal happy, and then anti-expressions. This was also true of older adults; the RTs and arousal ratings of the anti-expressions (i.e. anti-angry versus anti-happy) corresponded. However, this was not the case for normal happy expressions in older adults; these elicited the highest arousal ratings but were not detected most rapidly, which suggests that older adults indeed experienced emotional responses to happy facial expressions but that this was not directly reflected in the detection speed.

Older adults exhibited an impaired ability to detect happy faces despite preservation of the ability to detect angry faces. What are the theoretical implications of this? Older adults exhibit impaired recognition of negative facial expressions (including angry faces) but unimpaired recognition of happy emotions [43–47]. Thus, some authors have proposed that older adults tend to implement conscious emotional regulation strategies to continue to feel positive [7,42,44,46]. According to this theory, older adults recognize happy faces because they selectively seek them out. Similarly, older adults recognize

negative faces less well because they avoid them. Given that automatic detection of facial expressions is less likely to be influenced by a conscious emotional strategy in older adults [48], the apparently contrasting results between our study (on emotional detection) and previous studies (on emotional recognition) indicate that current theory should be expanded to more comprehensively consider automatic and conscious processing of the facial expressions.

Our results have also practical and clinical implications. Rapid detection of emotional facial expressions influences later stages of emotional processing, including processing of stimuli in social situations [32]. Sato *et al.* [32] found that compromised detection of smiling faces by people with autism spectrum disorder contributed to the difficulty these people experienced when creating affiliative interactions in social settings. In this context, it is possible that weakened automatic detection of happy faces among older adults would affect their day-to-day social activities; older adults may have difficulty detecting and focusing attention on smiling faces in their peripheral vision, given their poor performance detecting such expressions in our experiment, in which the faces were always presented peripherally. This has practical implications for caring for older patients; carers of older patients should consider the possibility that their patients might only poorly detect peripheral smiling faces. Clinical studies on the care of older patients have found that smiling with eye contact improves the quality of carer–cared interactions in caring settings [49–52]. Our results suggest that the beneficial effects of happy facial expressions may be reduced if carers are seen only peripherally by those for whom they care.

Our work has several limitations. We studied relationships between RTs and subjective ratings when facial presentation locations (central or peripheral) differed. In the future, it would be appropriate to present all faces peripherally in the two tasks; this would allow direct comparisons of the results. In addition, we were unable to create anti-expressions to emotional facial expressions featuring open mouths [6]; no stimulus revealed teeth. Happy facial expressions featuring open mouths might facilitate detection by older adults. In a related vein, we used only two models from the Ekman database, because all other models had open mouths [6]. These issues should be addressed in the future.

# 5. Conclusion

We found that older adults detect normal angry facial expressions more rapidly than anti-angry facial expressions, as do younger adults. This is the first report to show that older adults (like younger adults) detect angry facial expressions more rapidly than anti-expressions, which are typically recognized as neutral. Thus, rapid automatic detection of angry facial expressions remains unchanged throughout adulthood. Older adults (unlike younger adults) do not detect normal happy expressions more effectively than anti-happy expressions. This may be because happy expressions are less important to older adults.

Ethics. The ethics committee of the Unit for Advanced Studies of the Human Mind, Kyoto University approved this experiment. The experiment was also carried out following institutional ethical provisions and the Declaration of Helsinki. All of the participants gave written informed consent to participating in the experiment.

Data accessibility. The dataset of this article has been uploaded as supplementary material.

Authors' contributions. A.S. and W.S. designed the plan of this study; A.S. carried out the experiment; A.S. and W.S. analysed the data; A.S., W.S. and S.Y. wrote the manuscript.

Competing interests. We declare we have no competing financial interests.

Funding. Funds from the Japan Science and Technology Agency CREST (grant no. JPMJCR17A5) supported this study.

Acknowledgements. We are grateful to Yukari Sato for her technical support.

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
