## [Reviewer comments · Royal Society Open Science]

Review History

RSOS-191715.R0 (Original submission)

Review form: Reviewer 1

Is the manuscript scientifically sound in its present form?

No

Are the interpretations and conclusions justified by the results?

Yes

Is the language acceptable?

Yes

Do you have any ethical concerns with this paper?

No

Have you any concerns about statistical analyses in this paper?

No

Recommendation?

Major revision is needed (please make suggestions in comments)

Comments to the Author(s)

The current paper aims to examine the detection of happy and angry emotions (controlling for visual properties) in young and old adults. Normal and anti-expression of angry and happy faces were presented as a target among neutral distractors in a visual search task. Both age groups were faster to detect faces with normal angry expression compared to faces with anti-angry expression. Young adults have also detected faces with a normal happy expression. However, older adults did not show the detection advantage for happy over anti-happy expression. This suggests that older adults may face difficulties in terms of quick detection and focusing attention toward smiling faces.

Overall, I liked the research question and methodology. However, I have a few concerns, which need to be resolved before publications.

1. Authors have claimed that older adults have difficulty in detecting happy faces when it is not directly appearing in front of them. It implies that they do not have problems to detect a happy face when presented directly in front of them. Since they have not compared the detection of happy faces presented at the periphery compared to faces presented at the center of the screen, therefore, the authors need to be cautious in making such a conclusion. Alternatively, authors should find the previous literature where emotional faces were presented at the center of the screen, and they found the same results just like the current study. Authors can compare their results with previous studies to make such a claim.

2. The introduction is quite weak. For example, the authors were interested in comparing happy and angry emotions. However, in the introduction section, they have not mentioned the basic cognitive processing differences in the processing of happy and angry emotions (Gupta, 2019; Gupta et al., 2016). Also, a lot of literature is missing concerning emotion processing, specifically positive and negative emotion in general (Srinivasan & Gupta, 2010, 2011; Gupta & Srinivasan, 2015, Gupta & Deak, 2015). Please add these studies and discuss them. Moreover, authors have not mentioned what could be the possible cognitive-affective mechanisms underlying the processing of happy and angry expressions that would have resulted in differences in young and old adults to process emotional stimuli. Many studies have shown that very little attention is required to process pleasant stimuli compared to unpleasant stimuli (see Gupta, 2019, 2016; Gupta et al., 2016; Srinivasan & Gupta, 2010, 2011; Gupta & Srinivasan, 2015).

3. Authors have suggested that using anti-expression is more controlled stimuli compared to neutral stimuli. It is not very clear.

4. Please add some example of real faces (e.g., normal happy vs. anti happy face) used in the study.

5. It would be interesting to examine the detection RT/accuracy of emotional faces in the left vs. right visual field. It has been suggested that emotional face processing are right-hemisphere biased (see Gupta & Raymond, 2012; Gupta et al., 2018). This will give an insight into how emotions processing are represented in the old and young brain.

6. G-power analysis is required to determine the sample size.

7. Please address the theoretical and clinical implications of these results.

Gupta, R. (2019). Positive emotions have a unique capacity to capture attention. *Prog Brain Res*, 247:23-46. doi: 10.1016/bs.pbr.2019.02.001.

Gupta, R., Hur, Y., & Lavie, N. (2016). Distracted by pleasure?: Effects of positive versus negative valence on emotional capture under load. *Emotion*, 16(3), 328-337.

Srinivasan, N., & Gupta, G. (2010). Emotion – attention interactions in recognition memory for distractor faces. *Emotion*, 10, 207–215.

Srinivasan, N., & Gupta, R. (2011). Global-local processing affects recognition of distractor emotional faces. *Quarterly Journal of Experimental Psychology*, 64, 425-433.

Gupta, R., & Déak, G. O. (2015). Disarming smiles: irrelevant happy faces slow post-error responses. *Cognitive Processing*, 16, 427-434.

Gupta, R., & Srinivasan, N. (2015). Only irrelevant sad but not happy faces are inhibited under high perceptual load. *Cognition and Emotion*, 29, 747–754.

Gupta, R. & Raymond, J. E. (2012). Emotional distraction unbalances visual processing. *Psychonomic Bulletin and Review*, 19, 184-189.

Gupta, R., Raymond, J. E., & Vuilleumier, P. (2018). Priming by motivationally salient distractors produces hemispheric asymmetries in visual processing. *Psychological Research*. doi: 10.1007/s00426-018-1028-1.

Gupta, R. (2016) Commentary: Neural Control of Vascular Reactions: Impact of Emotion and Attention. *Front. Psychol.* 7:1613. doi: 10.3389/fpsyg.2016.01613.

Review form: Reviewer 2

Is the manuscript scientifically sound in its present form?

Yes

Are the interpretations and conclusions justified by the results?

No

Is the language acceptable?

Yes

Do you have any ethical concerns with this paper?

No

Have you any concerns about statistical analyses in this paper?

No

Recommendation?

Major revision is needed (please make suggestions in comments)

Comments to the Author(s)

In this behavioral study, Saito and colleagues compared the detection of angry and happy facial expressions (relative to neutral ones) between younger and older adults using a visual search task. The main finding is that older adults exhibited similar detection performance compared to younger ones in case of angry faces. However, the performance on happy faces differed between groups where older adults exhibited reduced detection performance. In general, the manuscript is well written, and the findings would be of interest to the affective neuroscience community. However, the authors should address the concerns noted below to clarify the design and interpretational issues further.

1. It is unclear why the authors had used only two facial stimuli though the Ekman face database contains stimuli from multiple actors? Because of this design choice, as the same target stimuli were repeated multiple (36) times across the experiment, could potential adaptation to happy faces in older adults explain the observed pattern of results??
2. In the RT Results section, it was reported that the older adults exhibited better performance on anti-happy compared to anti-angry faces. Why should this be the case, especially when the authors stated earlier that the anti faces were typically perceived as neutral facial expressions?
3. In the Discussion section, the arguments of linking reduced social reward processing in older adults to the observed pattern of results are very speculative as no measures of social reward processing were collected and compared between the two groups. The authors should at least explicitly acknowledge that such a speculative hypothesis could be explicitly tested in future work.
4. In the Rating task, were the facial stimuli presented centrally or peripherally (as in the visual search task)? If presented centrally, how meaningful would be to relate them to the performance with peripherally presented faces in the visual search task (especially in the older adults)??
5. It is a bit confusing to see the ANOVA findings based on the accuracy data under the Data Analyses section (page 15). It would be better if the authors could move them appropriately into the Results section.

Decision letter (RSOS-191715.R0)

12-Dec-2019

Dear Dr Saito,

The editors assigned to your paper ("Older Adults Detect Happy Facial Expressions Less Rapidly") have now received comments from reviewers. We would like you to revise your paper in accordance with the referee and Associate Editor suggestions which can be found below (not including confidential reports to the Editor). Please note this decision does not guarantee eventual acceptance.

Please submit a copy of your revised paper before 04-Jan-2020. Please note that the revision deadline will expire at 00.00am on this date. If we do not hear from you within this time then it will be assumed that the paper has been withdrawn. In exceptional circumstances, extensions may be possible if agreed with the Editorial Office in advance. We do not allow multiple rounds of revision so we urge you to make every effort to fully address all of the comments at this stage. If deemed necessary by the Editors, your manuscript will be sent back to one or more of the original reviewers for assessment. If the original reviewers are not available, we may invite new reviewers.

When submitting your revised manuscript, you must respond to the comments made by the referees and upload a file "Response to Referees" in "Section 6 - File Upload". Please use this to document how you have responded to the comments, and the adjustments you have made. In

order to expedite the processing of the revised manuscript, please be as specific as possible in your response.

- Data accessibility

If you wish to submit your supporting data or code to Dryad (<http://datadryad.org/>), or modify your current submission to dryad, please use the following link:
<http://datadryad.org/submit?journalID=RSOS&manu=RSOS-191715>

- Competing interests

- Authors' contributions

- Acknowledgements

- Funding statement

Kind regards,
Anita Kristiansen
Editorial Coordinator
Royal Society Open Science
openscience@royalsociety.org

on behalf of Dr Narayanan Srinivasan (Associate Editor) and Essi Viding (Subject Editor)
openscience@royalsociety.org

Associate Editor's comments (Dr Narayanan Srinivasan):

Two reviewers have now commented on the paper. Both find the paper interesting but have concerns. The authors need to address all the comments point by point in their revision.

Reviewers' Comments to Author:

Reviewer: 1

Comments to the Author(s)

The current paper aims to examine the detection of happy and angry emotions (controlling for visual properties) in young and old adults. Normal and anti-expression of angry and happy faces were presented as a target among neutral distractors in a visual search task. Both age groups were faster to detect faces with normal angry expression compared to faces with anti-angry expression. Young adults have also detected faces with a normal happy expression. However, older adults did not show the detection advantage for happy over anti-happy expression. This suggests that older adults may face difficulties in terms of quick detection and focusing attention toward smiling faces.

Overall, I liked the research question and methodology. However, I have a few concerns, which need to be resolved before publications.

1. Authors have claimed that older adults have difficulty in detecting happy faces when it is not directly appearing in front of them. It implies that they do not have problems to detect a happy face when presented directly in front of them. Since they have not compared the detection of happy faces presented at the periphery compared to faces presented at the center of the screen, therefore, the authors need to be cautious in making such a conclusion. Alternatively, authors should find the previous literature where emotional faces were presented at the center of the screen, and they found the same results just like the current study. Authors can compare their results with previous studies to make such a claim.

2. The introduction is quite weak. For example, the authors were interested in comparing happy and angry emotions. However, in the introduction section, they have not mentioned the basic cognitive processing differences in the processing of happy and angry emotions (Gupta, 2019; Gupta et al., 2016). Also, a lot of literature is missing concerning emotion processing, specifically positive and negative emotion in general (Srinivasan & Gupta, 2010, 2011; Gupta & Srinivasan, 2015, Gupta & Deak, 2015). Please add these studies and discuss them. Moreover, authors have not mentioned what could be the possible cognitive-affective mechanisms underlying the processing of happy and angry expressions that would have resulted in differences in young and old adults to process emotional stimuli. Many studies have shown that very little attention is required to process pleasant stimuli compared to unpleasant stimuli (see Gupta, 2019, 2016; Gupta et al., 2016; Srinivasan & Gupta, 2010, 2011; Gupta & Srinivasan, 2015).

3. Authors have suggested that using anti-expression is more controlled stimuli compared to neutral stimuli. It is not very clear.
4. Please add some example of real faces (e.g., normal happy vs. anti happy face) used in the study.
5. It would be interesting to examine the detection RT/accuracy of emotional faces in the left vs. right visual field. It has been suggested that emotional face processing are right-hemisphere biased (see Gupta & Raymond, 2012; Gupta et al., 2018). This will give an insight into how emotions processing are represented in the old and young brain.
6. G-power analysis is required to determine the sample size.
7. Please address the theoretical and clinical implications of these results.

Gupta, R. (2019). Positive emotions have a unique capacity to capture attention. *Prog Brain Res*, 247:23-46. doi: 10.1016/bs.pbr.2019.02.001.

Gupta, R., Hur, Y., & Lavie, N. (2016). Distracted by pleasure?: Effects of positive versus negative valence on emotional capture under load. *Emotion*, 16(3), 328-337.

Srinivasan, N., & Gupta, G. (2010). Emotion - attention interactions in recognition memory for distractor faces. *Emotion*, 10, 207-215.

Srinivasan, N., & Gupta, R. (2011). Global-local processing affects recognition of distractor emotional faces. *Quarterly Journal of Experimental Psychology*, 64, 425-433.

Gupta, R., & Déak, G. O. (2015). Disarming smiles: irrelevant happy faces slow post-error responses. *Cognitive Processing*, 16, 427-434.

Gupta, R., & Srinivasan, N. (2015). Only irrelevant sad but not happy faces are inhibited under high perceptual load. *Cognition and Emotion*, 29, 747-754.

Gupta, R. & Raymond, J. E. (2012). Emotional distraction unbalances visual processing. *Psychonomic Bulletin and Review*, 19, 184-189.

Gupta, R., Raymond, J. E., & Vuilleumier, P. (2018). Priming by motivationally salient distractors produces hemispheric asymmetries in visual processing. *Psychological Research*. doi: 10.1007/s00426-018-1028-1.

Gupta, R. (2016) Commentary: Neural Control of Vascular Reactions: Impact of Emotion and Attention. *Front. Psychol.* 7:1613. doi: 10.3389/fpsyg.2016.01613.

Reviewer: 2

Comments to the Author(s)

In this behavioral study, Saito and colleagues compared the detection of angry and happy facial expressions (relative to neutral ones) between younger and older adults using a visual search task. The main finding is that older adults exhibited similar detection performance compared to younger ones in case of angry faces. However, the performance on happy faces differed between groups where older adults exhibited reduced detection performance. In general, the manuscript is well written, and the findings would be of interest to the affective neuroscience community. However, the authors should address the concerns noted below to clarify the design and interpretational issues further.

1. It is unclear why the authors had used only two facial stimuli though the Ekman face database contains stimuli from multiple actors? Because of this design choice, as the same target stimuli were repeated multiple (36) times across the experiment, could potential adaptation to happy faces in older adults explain the observed pattern of results??
2. In the RT Results section, it was reported that the older adults exhibited better performance on anti-happy compared to anti-angry faces. Why should this be the case, especially when the authors stated earlier that the anti faces were typically perceived as neutral facial expressions?
3. In the Discussion section, the arguments of linking reduced social reward processing in older adults to the observed pattern of results are very speculative as no measures of social reward processing were collected and compared between the two groups. The authors should at least explicitly acknowledge that such a speculative hypothesis could be explicitly tested in future work.
4. In the Rating task, were the facial stimuli presented centrally or peripherally (as in the visual search task)? If presented centrally, how meaningful would be to relate them to the performance with peripherally presented faces in the visual search task (especially in the older adults)??
5. It is a bit confusing to see the ANOVA findings based on the accuracy data under the Data Analyses section (page 15). It would be better if the authors could move them appropriately into the Results section.

Author's Response to Decision Letter for (RSOS-191715.R0)

See Appendix A.

RSOS-191715.R1 (Revision)

Review form: Reviewer 1

Is the manuscript scientifically sound in its present form?

Yes

Are the interpretations and conclusions justified by the results?

Yes

Is the language acceptable?

Yes

Do you have any ethical concerns with this paper?

No

Have you any concerns about statistical analyses in this paper?

No

Recommendation?

Accept as is

Comments to the Author(s)

Authors have incorporated all comments.

Review form: Reviewer 2

Is the manuscript scientifically sound in its present form?

Yes

Are the interpretations and conclusions justified by the results?

Yes

Is the language acceptable?

Yes

Do you have any ethical concerns with this paper?

No

Have you any concerns about statistical analyses in this paper?

No

Recommendation?

Accept as is

Comments to the Author(s)

The authors have done a great job in addressing my previous comments. I do not have any additional comments or concerns.

Decision letter (RSOS-191715.R1)

21-Feb-2020

Dear Dr Saito,

It is a pleasure to accept your manuscript entitled "Older Adults Detect Happy Facial Expressions Less Rapidly" in its current form for publication in Royal Society Open Science. The comments of the reviewer(s) who reviewed your manuscript are included at the foot of this letter.

You can expect to receive a proof of your article in the near future. Please contact the editorial office (openscience_proofs@royalsociety.org) and the production office (openscience@royalsociety.org) to let us know if you are likely to be away from e-mail contact -- if

you are going to be away, please nominate a co-author (if available) to manage the proofing process, and ensure they are copied into your email to the journal.

on behalf of Dr Narayanan Srinivasan (Associate Editor) and Essi Viding (Subject Editor)
openscience@royalsociety.org

Associate Editor Comments to Author (Dr Narayanan Srinivasan):

Associate Editor: 1

Comments to the Author:

Both reviewers are happy with the revisions. Congratulations and the paper is now accepted for publication in RSOS.

Reviewer comments to Author:

Reviewer: 1

Comments to the Author(s)

Authors have incorporated all comments.

Reviewer: 2

Comments to the Author(s)

The authors have done a great job in addressing my previous comments. I do not have any additional comments or concerns.

Appendix A

January 04, 2020

Dear Dr. Narayanan Srinivasan and Essi Viding

We have uploaded the revised manuscript of “Older adults detect happy facial expressions less rapidly.” We would like to thank the Associate Editor and the reviewers for reviewing our manuscript and giving us invaluable comments. We have revised the manuscript following your comments and major changes are highlighted. Additionally a professional English-language editing service made language-related changes, which are not highlighted unless the scientific content was altered.

We believe that the revised manuscript has improved thanks to your comments and suggestions. We would be grateful if you would consider this revised manuscript for publication in *Royal Society Open Science*.

Reviewer #1

Comment 1

Authors have claimed that older adults have difficulty in detecting happy faces when it is not directly appearing in front of them. It implies that they do not have problems to detect a happy face when presented directly in front of them. Since they have not compared the detection of happy faces presented at the periphery compared to faces presented at the center of the screen, therefore, the authors need to be cautious in making such a conclusion. Alternatively, authors should find the previous literature where emotional faces were presented at the center of the screen, and they found the same results just like the current study. Authors can compare their results with previous studies to make such a claim.

Response

We agree with the reviewer on this point. Our results suggest that older adults may have difficulty detecting happy faces peripherally, therefore we have denoted this result and deleted the speculative expressions (in Abstract, Discussion, and Conclusion).

Comment 2

The introduction is quite weak. For example, the authors were interested in comparing happy and angry emotions. However, in the introduction section, they have not mentioned the basic cognitive processing differences in the processing of happy and angry emotions (Gupta, 2019; Gupta et al., 2016). Also, a lot of literature is missing concerning emotion processing, specifically positive and negative emotion in general (Srinivasan & Gupta, 2010, 2011; Gupta & Srinivasan, 2015, Gupta & Deak, 2015). Please add these studies and discuss them. Moreover, authors have not mentioned what could be the possible cognitive-affective mechanisms underlying the processing of happy and angry expressions that would have resulted in differences in young and old adults to process emotional stimuli. Many studies have shown that very little attention is required to process pleasant stimuli compared to

unpleasant stimuli (see Gupta, 2019, 2016; Gupta et al., 2016; Srinivasan & Gupta, 2010, 2011; Gupta & Srinivasan, 2015).

Response

We have revised the Introduction section to cite studies that have examined the capturing power of positive (happy) stimuli (pages 5–6 in Introduction). Additionally, we have discussed the significance of positive stimuli on human beings and have interpreted our results as reflecting the decreased value of happy faces for older adults (page 18 in Discussion).

Comment 3

Authors have suggested that using anti-expression is more controlled stimuli compared to neutral stimuli. It is not very clear.

Response

Anti-expressions are not only generally rated as neutral expressions, but are also controlled stimuli for normal expressions in terms of visual (physical) characteristics of faces. In the previous version of the manuscript, we mentioned anti-expressions but failed to fully explain the term. In the revised manuscript, we have added the necessary explanation (page 3).

Comment 4

Please add some example of real faces (e.g., normal happy vs. anti happy face) used in the study.

Response

We cannot show real faces due to the lack of a license. Although all images published at Royal Society Open Science must be under the Creative Commons Attribution (CC BY) license (<https://royalsociety.org/journals/authors/author-guidelines>), the copyright for the images used in our study is held by The Paul Ekman Group, LLC and each reprint requires separate permission and payment. Therefore, we modified Figure 1 to show more realistic illustrations.

Comment 5

It would be interesting to examine the detection RT/accuracy of emotional faces in the left vs. right visual field. It has been suggested that emotional face processing are right-hemisphere biased (see Gupta & Raymond, 2012; Gupta et al., 2018). This will give an insight into how emotions processing are represented in the old and young brain.

Response

We conducted an ANOVA by adding the factor of visual field (right field versus left field). Because the result did not show a significant four-way interaction ($F(1,60) = 1.431, p > .10$), we reported it as the result of preliminary analysis in the Materials and Methods section (page 13).

Comment 6

G-power analysis is required to determine the sample size.

Response

We have performed G-power analysis and described it in the Materials and Method section (pages 7–8).

Comment 7

Please address the theoretical and clinical implications of these results.

Response

We have discussed the implications of our results (pages 19–20 in Discussion).

Reviewer #2**Comment 1**

It is unclear why the authors had used only two facial stimuli though the Ekman face database contains stimuli from multiple actors? Because of this design choice, as the same target stimuli were repeated multiple (36) times across the experiment, could potential adaptation to happy faces in older adults explain the observed pattern of results??

Response

The main reason that we did not use faces from other models is because it was not possible to create corresponding anti-expressions of anger and happiness from other models, particularly those with open mouths. We have made a reference to this limitation (pages 20–21 in Discussion).

Regarding the issue of the potential adaptation to happy faces, we examined whether adaptation effects would be observed. We conducted an ANOVA by adding the factor of presentation block. Because there was no significant four-way interaction ($F(3,180) = 1.175, p > .10$), we reported it as the result of preliminary analysis in the Materials and Methods section (page 13).

Comment 2

In the RT Results section, it was reported that the older adults exhibited better performance on anti-happy compared to anti-angry faces. Why should this be the case, especially when the authors stated earlier that the anti faces were typically perceived as neutral facial expressions?

Response

We have discussed this pattern of results by comparing them to the rating performance of these stimuli (pages 18–19 in Discussion).

Comment 3

In the Discussion section, the arguments of linking reduced social reward processing in older adults to the observed pattern of results are very speculative as no measures of social reward

processing were collected and compared between the two groups. The authors should at least explicitly acknowledge that such a speculative hypothesis could be explicitly tested in future work.

Response

We did not conduct any tests related to social reward processing. Instead, we have attempted to explain our results using the hypotheses from studies that have examined the interaction between positive stimuli (including happy faces) and attention (page 18).

Comment 4

In the Rating task, were the facial stimuli presented centrally or peripherally (as in the visual search task)? If presented centrally, how meaningful would be to relate them to the performance with peripherally presented faces in the visual search task (especially in the older adults)??

Response

We have acknowledged that this is one of the limitations of our study and we have mentioned that presenting the faces in the same manner across the tasks would be necessary to examine the relationship between the performances of these tasks (page 20 in Discussion).

Comment 5

It is a bit confusing to see the ANOVA findings based on the accuracy data under the Data Analyses section (page 15). It would be better if the authors could move them appropriately into the Results section.

Response

We have updated the manuscript accordingly (page 15).